# Leukoaraiosis as a Predictor of Depression and Cognitive Impairment among Stroke Survivors: A Systematic Review

**Eftychia Tziaka** [1,†] 🄳, **Foteini Christidi** [1,†], **Dimitrios Tsiptsios** [1,*] 🄳, **Anastasia Sousanidou** [1] 🄳, **Stella Karatzetzou** [1], **Anna Tsiakiri** [1] 🄳, **Triantafyllos K. Doskas** [1], **Konstantinos Tsamakis** [2] 🄳, **Nikolaos Retzepis** [3], **Christos Konstantinidis** [3], **Christos Kokkotis** [3], **Aspasia Serdari** [4], **Nikolaos Aggelousis** [3] 🄳 and **Konstantinos Vadikolias** [1] 🄳

1  Neurology Department, Democritus University of Thrace, 68100 Alexandroupolis, Greece
2  Institute of Psychiatry, Psychology and Neuroscience (IoPPN), King's College London, London SE5 8AB, UK
3  Department of Physical Education and Sport Science, Democritus University of Thrace, 69100 Komotini, Greece
4  Department of Child and Adolescent Psychiatry, Medical School, Democritus University of Thrace, 68100 Alexandroupolis, Greece
*  Correspondence: tsiptsios.dimitrios@yahoo.gr; Tel.: +30-6944320016
†  These authors contributed equally to this work.

**Abstract:** Stroke survivors are at increased risk of developing depression and cognitive decline. Thus, it is crucial for both clinicians and stroke survivors to be provided with timely and accurate prognostication of post-stroke depression (PSD) and post-stroke dementia (PSDem). Several biomarkers regarding stroke patients' propensity to develop PSD and PSDem have been implemented so far, leukoaraiosis (LA) being among them. The purpose of the present study was to review all available work published within the last decade dealing with pre-existing LA as a predictor of depression (PSD) and cognitive dysfunction (cognitive impairment or PSDem) in stroke patients. A literature search of two databases (MEDLINE and Scopus) was conducted to identify all relevant studies published between 1 January 2012 and 25 June 2022 that dealt with the clinical utility of preexisting LA as a prognostic indicator of PSD and PSDem/cognitive impairment. Only full-text articles published in the English language were included. Thirty-four articles were traced and are included in the present review. LA burden, serving as a surrogate marker of "brain frailty" among stroke patients, appears to be able to offer significant information about the possibility of developing PSD or cognitive dysfunction. Determining the extent of pre-existing white matter abnormalities can properly guide decision making in acute stroke settings, as a greater degree of such lesioning is usually coupled with neuropsychiatric aftermaths, such as PSD and PSDem.

**Keywords:** leukoaraiosis; white matter hyperintensities; stroke; depression; dementia; cognitive impairment

## 1. Introduction

Globally, strokes pose a huge challenge for healthcare. Even though significant progress has been made regarding early diagnosis and treatment, stroke is still the second leading cause of death worldwide, following heart disease, and the leading cause of acquired disability in adults [1,2]. Considering stroke's age-related nature, with more than 50% of patients being over 65 [3], and the ever-expanding lifespans of individuals, which will almost triple the number of adults over 60 in developed countries by 2050 (United Nations DoEaSA Population Division, 2007), it is expected that stroke survivors will increase dramatically in the future. Thus, a crucial need for prompt and accurate identification of patients with an unfavorable prognosis has emerged. Additionally, such an approach may allow the development of personalized rehabilitation programs based on the propensity for recovery of each individual [4].

In the neuropsychiatric aftermath of a stroke, depression (post-stroke depression (PSD)) is the most common and burdensome outcome [5,6]. PSD is defined according to the Diagnostic and Statistical Manual of Mental Disorders (DSM-5) criteria as a *depressive disorder due to another medical condition* (i.e., *stroke*). The diagnosis is made on the basis of five criteria: (A) the presence of depressed mood or anhedonia; (B) the symptoms are pathophysiologically related to the stroke; (C) other psychiatric disorders do not better explain or describe the symptoms; (D) disturbance does not occur exclusively in the presence of delirium; and (E) symptoms cause significant distress or impairment to the patient. PSD can be further specified as *a depressive disorder due to stroke* when the patient has a depressed mood and/or anhedonia but does not meet the full criteria for major depressive disorder (MDD). However, it can also be defined as a depressive disorder when the patient goes through a major depression-like episode (when the full criteria for MDD are met; i.e., five of nine MDD criteria for at least two weeks). Finally, the patient may exhibit mixed features (i.e., symptoms of hypomania or mania may also be present but do not predominate the clinical picture).

In terms of its epidemiology, meta-analyses estimate the cross-sectional prevalence of PSD as between 18 and 33% [7–10]. Furthermore, according to a meta-analysis of longitudinal studies, 55% of post-stroke patients are depressed at some point after the vascular event. Additionally, people with PSD have higher mortality rates [9,11,12], more pronounced cognitive deficits [13], higher long-term disability odds ratios [14], lower quality of life [15], and higher rates of suicidal ideation and attempts [16] as compared to post-stroke patients without depression. It is thus critical to understand the principles of identification and effective treatment options for PSD.

Stroke survivors are also at increased risk of developing cognitive impairments, as the acute tissue damage may affect their cognitive status [17]. Therefore, for the diagnosis of post-stroke dementia (PSDem), memory impairment is an essential feature in addition to the vascular event. Additionally, a decline in one or more cognitive domains is required (e.g., executive function, memory, language, visuospatial and constructional dexterities), and the loss of function has to be severe enough to interfere with the individual's social or occupational functioning [18].

Epidemiologically, PSDem ranks second among causes of cognitive decline, behind only Alzheimer's disease (AD) [19]. Moreover, the lifetime odds of developing either stroke or dementia at the age of 65 are one in three for men and one in two for women [20]. The absolute number of people suffering from PSDem will rise as a result of changing demographics, increased life expectancy, and improved stroke survival rates. Nevertheless, due to its relation to stroke incidence, PSDem could be reduced with improved stroke prevention [17]. However, despite the availability of prospective data, conflicting results have been reported, and it remains unclear whether strokes directly affect cognitive function beyond vascular risk factors and age-related cognitive decline [17].

Due to the high incidence of stroke in the general population, several techniques and assessment tools have been developed for the prevention and early detection of factors that may lead to stroke, its recurrence, or the occurrence of associated aggravating symptoms [21]. A key point in this prevention effort is the use of biomarkers. Despite their primary roles diagnosing disease and predicting outcomes, biomarkers in stroke patients can also be utilized to provide a wide range of other information about future stroke risks, possible mechanisms of stroke that can be used to guide treatment, and drug responses [22]. Furthermore, due to the heterogeneity of strokes, biomarkers can provide personalized information based on the needs of the patient, making it possible to speed up recovery, prevent associated physical and cognitive changes, follow an effective medication line, and ensure the best possible living conditions [22–25].

To date, several biomarkers have been identified, such as serum biomarkers, genetic markers, and cerebrospinal fluid biomarkers, that can help in identifying signs of depression and cognitive decline after stroke. These biomarkers assist medical professionals in constructing specialized and effective treatments for their patients and developing

prevention protocols [17]. One of the biomarkers frequently reported in stroke cases is leukoaraiosis (LA), an indicator also known as white matter lesions (WMLs). Through neuroimaging, specific abnormalities in the brain's white matter can be observed. These abnormalities appear either as polyfocal or widespread changes of varying sizes, and they are mainly located in the periventricular cavity and often detected in older adults [21]. LA is described differently depending on the method with which it is approached and identified. For example, it has been described as hypodense brain areas in computed tomography (CT) scans and as white matter hyperintensities (WMHs) in T2-weighted (T2WI) and/or fluid-attenuated inversion recovery (FLAIR) magnetic resonance imaging (MRI) sequences [26].

The association of LA with stroke and its consequences is a significant and interesting issue in the current literature [27,28]. LA is an independent predictor of stroke [29], hemorrhagic transformation following thrombolysis for ischemic stroke [30], and PSDem [31,32]. LA progression occurs across the board in older people with moderate-to-severe white matter changes, regardless of their clinical symptoms [33].

It is now widely known that LA is associated with cognitive dysfunction and a depressive symptomatology [34,35]. Following a stroke, LA may contribute to the development of new cognitive deficits and late-onset depression [36]. Regarding the severity of LA and psychiatric disorders, the progression of white matter abnormalities has been found to be causally associated with depressive symptoms, indicating and highlighting the crucial role LA plays in the development of depression later in life [37].

Furthermore, LA burden has a negative impact on specific cognitive domains, such as executive function, attention, and processing speed, as well as leading to an accelerated decline in overall cognitive performance [38,39]. In addition to cognitive decline, an increasing number of white matter lesions can potentially predict the development of mild cognitive impairment, dementia, and disability [38]. After one year of follow-up, it was found that individuals with severe white matter changes had poorer overall cognitive performance and exhibited rapid emotional and functional decline compared to those with mild LA at baseline [38,40].

Thus, LA may affect mood status and cognition after stroke through similar pathophysiological mechanisms. Therefore, delving into and further studying this indicator and its predictive functions will provide helpful and insightful information that can assist in preventing and controlling such conditions effectively. Such actions will enable the medical community to recognize and rapidly treat the adverse and harmful effects PSD and PSDem can have on patients.

Taking into consideration the clinical relevance and the potential prognostic role of baseline and severe LA within an aging population, as well as the emerging need for accurate forecasting of each stroke individual's propensity for depression, cognitive impairment, and recovery, the purpose of the present study was to review all available work published within the last decade dealing with pre-existing LA as a predictor of depression (PSD) and cognitive dysfunction (cognitive impairment or PSDem) in stroke patients.

## 2. Materials and Methods

The Preferred Reporting Items for Systematic Reviews and Meta-analyses (PRISMA registration number: CRD42023388973) were used to guide this study. Our study's methods were designed a priori.

### 2.1. Search Strategy

Two investigators (AS and DT) conducted a literature search of two databases (MEDLINE and Scopus) to trace all relevant studies published between 1 January 2012 and 25 June 2022. Search terms were as follows: ("leukoaraiosis" OR "white matter hyperintensities" OR "WMHs" OR "small vessel disease") AND ("poststroke" OR "stroke outcome") AND ("cognition" OR "dementia" OR "depression"). Table 1 lists the structured search techniques for each database. The retrieved articles were also manually searched for any fur-

ther potentially eligible articles. Any disagreement regarding the screening or the selection process was discussed with a third investigator (KV) until a consensus was reached.

**Table 1.** Employed databases and search strategies.

| Database | Search Strategy |
|---|---|
| PubMed | ("leukoaraiosis" OR "white matter hyperintensities" OR "WMHs" OR "small vessel disease") AND ("poststroke" OR "stroke outcome") AND ("cognition" OR "dementia" OR "depression") |
| Scopus | ("leukoaraiosis" OR "white matter hyperintensities" OR "WMHs" OR "small vessel disease") AND ("poststroke" OR "stroke outcome") AND ("cognition" OR "dementia" OR "depression") |

Note. WHMs = white matter hyperintensities.

*2.2. Selection Criteria*

Only original full-text articles published in the English language were included. Secondary analyses, reviews, guidelines, meeting summaries, comments, unpublished abstracts, and studies conducted with animals were excluded. There were no restrictions on the study design or sample characteristics.

*2.3. Data Extraction*

Data extraction was performed using a predefined data form created in Excel. We recorded the type of stroke and specific ischemic stroke-related details (when available), the authors, the year of publication, the type of study, the number of participants, demographic (age, gender, education) and other socio-economic data (marital/occupational status), the body mass index (BMI), the cerebrovascular risk factors mentioned, whether a previous stroke was referenced, the follow-up time, the time of MRI acquisition and method of LA assessment, the clinical and psychometric (cognitive or mood-related) scales used, and the main findings.

*2.4. Data Analysis*

We performed quality assessment of included studies using the Newcastle–Ottawa Scale (NOS) for observational studies (www.ohri.ca) (accessed on 28 December 2022). Modified versions of the NOS (mNOS) have been widely used in the literature, and we selected the mNOS used in a previous study on the prevalence of dementia in ischemic- and mixed-stroke patients [41]. Coding modifications regarding dementia were adjusted for depression and cognitive impairment/dementia in our study. Based on the mNOS, we rated seven items classified into three categories: selection of study groups, comparability of the groups, and ascertainment of outcome of interest. Each study was awarded a maximum of one point for each numbered item within the selection and outcome categories, while a maximum of two points were given for the comparability category. The final mNOS score ranged between 0 and 8, with higher scores corresponding to higher quality studies. As no formal cut-off values were available to define high or low risk of bias, we applied previously used cut-off values [41]. Scores of 6–8, 3–5, and 0–2 points represented high, moderate, and low quality, respectively. No statistical analysis or meta-analysis was performed due to the high level of heterogeneity among the studies. Thus, the data were only descriptively analyzed.

**3. Results**

*3.1. Database Searches and Quality Assessment of Included Studies*

Overall, 631 records were retrieved from the database search. Duplicates and irrelevant studies were excluded; hence, a total of 124 articles were selected. After screening the full texts of the articles, 34 studies were judged to be eligible for inclusion (Figure 1). Thirty-two studies were categorized as high quality and two studies were categorized as moderate quality.

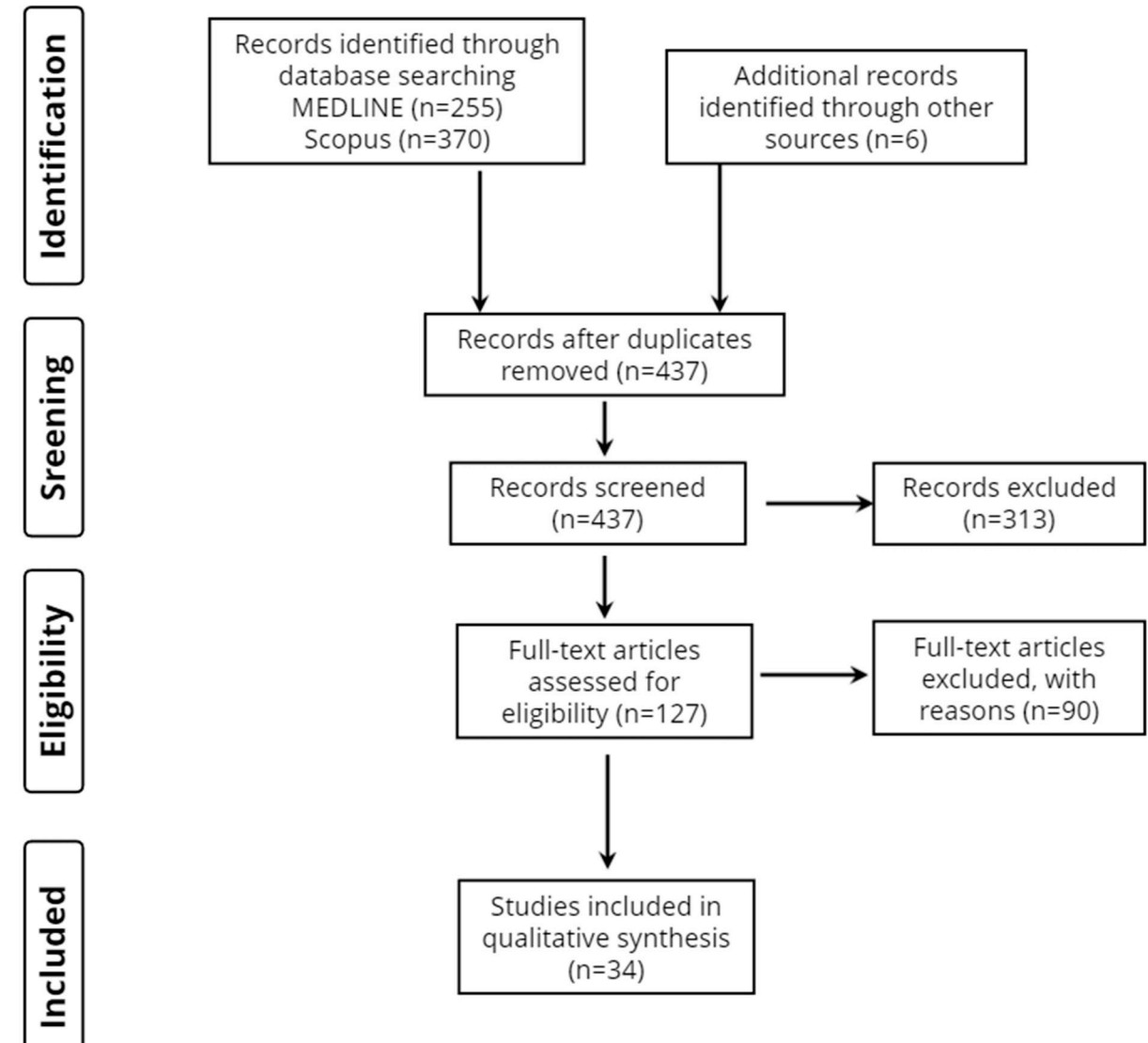

**Figure 1.** Study flow diagram (PRISMA flowchart).

*3.2. Study Characteristics*

Thirty-four publications met the inclusion criteria and are presented in detail in Tables 2–4 based on their main focus (i.e., depression (n = 8 studies), cognition (n = 25 studies), and depression and cognition (n = 1 study)). Additional details regarding patients' clinical characteristics, as well as the mNOS score of each study, are presented in Supplementary Tables S1–S3. Thirteen studies focused entirely on acute ischemic stroke (AIS), eleven studied patients with any stroke type, seven included patients with either AIS or transient ischemic attacks (TIA), and three enrolled intracerebral hemorrhage patients. With regard to the origin of the studies, 18 were from Asia, 13 from Europe, and 3 from America.

**Table 2.** Characteristics of the included studies focusing on depression.

| First Author (Year) | Type of Stroke, Study Design, Follow-Up Time, Participants (n) | Patients' Demographics: Age (Years), Gender (M/F), Education (Years/Level), Marital/Occupational Status, Income, BMI | Time of MRI Acquisition/Leukoaraiosis or WHM Assessment | Clinical and/or Psychometric Scales | Main Results |
|---|---|---|---|---|---|
| 1. Guo (2022) [42] | • AIS<br>• Longitudinal<br>• 1 week<br>• 372 patients | • Age: 58.6 ± 12<br>• 255 M/117 F<br>• Education: n/a<br>• BMI 25.0 ± 3.3 | • Within 7 days of admission<br>• Fazekas scale | NIHSS on admission and HAMD-24 and the Chinese version of the Lubben Social Network Scale one week post-stroke | • Total SVD scores (which included deep and periventricular WMH scores) were significantly associated with PSD, even after adjusting for age and gender |
| 2. Jaroonpipatkul (2022) [43] | • AIS<br>• Longitudinal<br>• 3 months<br>• 43 patients/17 controls | • Age: NIHSS < 3—63.1 ± 6.7, NIHSS ≥ 3—65.2 ± 8.6<br>• 33 M/27 F<br>• Education: NIHSS < 3—10.8, NIHSS ≥ 3—7.9 | • At baseline<br>• Volumetric | NIHSS at baseline and at 3 months, mRS at baseline, and Montgomery–Åsberg Depression Rating Scale at 3 months | • Total WMHs had significant specific indirect effects on key depressive symptoms, concentration–tension symptoms, and lassitude, and the latter effects were mediated in different paths via left or right infarct sizes and accompanying disabilities |
| 3. Zhou (2022) [44] | • AIS<br>• Longitudinal<br>• 14 ± 2 days<br>• 346 patients | • Age: non-PSD—63 (53–70), PSD—64 (59–68)<br>• 224 M/122 F<br>• Education: illiterate—n = 29, primary school—n = 194, middle school and higher—n = 123 | • n/a<br>• Fazekas scale | NIHSS and BI at baseline and MMSE and HAMD-17 at follow-up | • Mild-to-severe WMH scores were different between the non-PSD and PSD groups<br>• A range of 2–4 points for the total cSVD burden was an independent risk factor for early-onset PSD |

**Table 2.** *Cont.*

| | First Author (Year) | Type of Stroke, Study Design, Follow-Up Time, Participants (n) | Patients' Demographics: Age (Years), Gender (M/F), Education (Years/Level), Marital/Occupational Status, Income, BMI | Time of MRI Acquisition/Leukoaraiosis or WHM Assessment | Clinical and/or Psychometric Scales | Main Results |
|---|---|---|---|---|---|---|
| 4. | Douven (2020) [45] | • Any type (ischemic n = 176, hemorrhagic n = 12)<br>• Longitudinal<br>• 12 months<br>• 188 patients | • Age: 65.4 ± 11.1<br>• 132 M/56 F<br>• Education: low—n = 68, middle—n = 70, high—n = 49 | • At 3 months<br>• Volumetric | MINI and Apathy Evaluation Scale | • Patients with post-stroke apathy had higher WMH volume<br>• Severe cSVD burden was significantly associated with higher odds of post-stroke apathy and post-stroke depression |
| 5. | Bae (2019) [46] | • AIS<br>• Longitudinal<br>• 2 weeks to 1 year<br>• 408 patients | • Age: 64.7 ± 10.0<br>• 237 M/171 F<br>• Education: PSD group—7.8, non-PSD group—8.8 | • On admission<br>• Fazekas scale | NIHSS, BI, MMSE, and MINI on admission and at 1 year follow-up | • Severe periventricular WMHs were significantly associated with any type of post-stroke depression at 2 weeks and severe deep WMHs were significantly associated with major post-stroke depression at 1 year follow-up, even after adjusting for age, sex, education, previous history of depression and stroke, NIHSS and MMSE scores, and stroke location |

**Table 2.** *Cont.*

| First Author (Year) | Type of Stroke, Study Design, Follow-Up Time, Participants (n) | Patients' Demographics: Age (Years), Gender (M/F), Education (Years/Level), Marital/Occupational Status, Income, BMI | Time of MRI Acquisition/Leukoaraiosis or WHM Assessment | Clinical and/or Psychometric Scales | Main Results |
|---|---|---|---|---|---|
| 6. Carnes-Vendrell (2019) [47] | • TIA or minor stroke (NIHSS ≤ 4)<br>• Longitudinal<br>• 12 months<br>• 82 patients | • Age: 66.4 ± 11.0<br>• 59 M/23 F,<br>• Education: primary—n = 57, secondary or more—n = 25<br>• Married—n = 63, widower—n = 11, divorced/separated—n = 6, single—n = 2,<br>• Employed:—n = 22, unemployed—n = 4, retired—n = 51, unable to work—n = 5 | • Within < 7 days<br>• Fazekas scale | NIHSS, BI, and mRS at baseline and the Beck Depression Inventory and Montgomery–Åsberg Depression Rating Scale | • Fazekas hyperintensity in the deep white matter substance score was related to post-stroke depression at 12 months, and the Fazekas periventricular score was associated with basal poststroke apathy |
| 7. Pavlovic (2016) [48] | • Small subcortical stroke of lacunar type<br>• Longitudinal<br>• 3–5 years<br>• 294 patients | • Age: 62.3 ± 10.7<br>• 158 M/136 F<br>• Education: 11.8 ± 2.2 | • n/a<br>• ARWMC, Fazekas scale, Wahlund scale | mRS | • Total ARWMC score was an independent predictor of post-stroke depression |
| 8. Tanislav (2015) [49] | • Any type (TIA, Ischemic stroke, primary hemorrhage, other)<br>• Longitudinal<br>• 3 months<br>• 2007 patients | • Age: 46 (IQR 40–51)<br>• 1151 M/856 F<br>• Education: n/a | • Within ≤ 1 month of inclusion<br>• Fazekas scale | NIHSS at baseline and BDI | • No significant associations between clinically relevant depressive symptoms and cerebral MRI findings, such as the presence of acute infarcts, old infarctions, or WMHs, were found |

Notes. n/a = not available; N = number; M/F = male/female; MRI = magnetic resonance imaging; WHMs = white matter hyperintensities; AIS = acute ischemic stroke; PSD = post-stroke depression; NIHSS = National Institutes of Health Stroke Scale; BI = Barthel Index; MMSE = Mini-Mental State Examination; MINI = Mini International Neuropsychiatric Interview; ARWMC = Age-Related White Matter Change; mRS = Modified Rankin Scale; TIA = transient ischemic attack; cSVD = cerebral small vessel disease; HAMD = Hamilton Depression Scale; SVD = small vessel disease; BMI = body mass index; BDI = Beck Depression Inventory; mNOS = Modified Newcastle–Ottawa Scale.

**Table 3.** Characteristics of included studies focusing on cognition.

| First Author (Year) | Type of Stroke, Study Design, Follow-Up Time, Participants (n) | Patients' Demographics: Age (Years), Gender (M/F), Education (Years/Level), Marital/Occupational Status, Income, BMI | Time of MRI Acquisition/Leukoaraiosis or WHM Assessment | Clinical and/or Psychometric Scales | Main Results |
|---|---|---|---|---|---|
| 1. Georgakis (2022) [49] | • Any type<br>• Longitudinal<br>• 12 months<br>• 666 patients | • Age: 67.9 ± 11.4<br>• 444 M/222 F<br>• Education: 13 (IQR 12–16)<br>• BMI: 27.0 ± 4.3 | • Within 5 days post-stroke<br>• Fazekas scale | NIHSS, mRS, GCS, MMSE, and MoCA at baseline and mRS, a global functional scale focused on motor recovery, BI, and the IADLs at follow-up | • Both deep and periventricular WMH grades showed significant associations with worse outcomes for global cognition, executive function, attention, visuospatial ability, and functional status |
| 2. Fruhwirth (2021) [50] | • Small subcortical infarction<br>• Longitudinal<br>• 15 months<br>• 82 patients | • Age: 61 ± 10<br>• 73 M/19 F<br>• Education: 12 ± 3 | • At baseline and 15 months<br>• Volumetric, Fazekas scale | NIHSS score and mRS | • Although patients without or with only mild WMHs improved in set-shifting after 15 months, there were no improvements in patients with moderate-to-severe WMHs.<br>• Baseline total WMH volume was the only significant predictor for attention 15 months post-stroke |
| 3. Pasi (2021) [51] | • ICH<br>• Longitudinal<br>• Median of 46.3 months<br>• 612 patients | • Age: 70.5<br>• 324 M/288 F<br>• Education (≥12 y): n = 361 | • Within 90 days of symptoms' onset<br>• Fazekas scale | NIHSS and IQCODE at baseline | • Grade 3 periventricular WMHs were associated with increasing cerebral atrophy scores<br>• Periventricular WMHs were not associated with individual rates of cognitive decline.<br>• Severe deep WMHs were linked to dementia risk after ICH |

**Table 3.** *Cont.*

| | First Author (Year) | Type of Stroke, Study Design, Follow-Up Time, Participants (n) | Patients' Demographics: Age (Years), Gender (M/F), Education (Years/Level), Marital/Occupational Status, Income, BMI | Time of MRI Acquisition/Leukoaraiosis or WHM Assessment | Clinical and/or Psychometric Scales | Main Results |
|---|---|---|---|---|---|---|
| 4. | Peng (2021) [52] | • AIS<br>• Longitudinal<br>• Until rehabilitation discharge<br>• 144 patients/ 30 controls | • Age: patients with lower FIM cognitive scores on admission—69.43 ± 10.71, patients with high FIM cognitive scores on admission—68.93 ± 11.24<br>• 92 M/52 F<br>• Education: n/a<br>• BMI: patients with low FIM cognitive scores on admission—27.30 ± 5.69, patients with high FIM cognitive scores on admission—28.59 ± 5.86 | • Within 1 month post-stroke<br>• Volumetric | NIHSS, mRS, and FIM cognitive sub-score | • WMH volume was negatively associated with FIM cognitive sub-scores on admission<br>• The combination of serum neurofilament light chain (NfL) levels and volumes of infarcts and WMHs showed an improved predictive value for cognitive function during the acute rehabilitation phase after stroke |
| 5. | Sung (2021) [53] | • AIS<br>• Longitudinal<br>• 1 year<br>• 112 patients | • Age: 64.5 (IQR 57–73.5)<br>• 72 M/40 F<br>• Education: 9.0 (IQR 6.0–12.0) | • Within 7 days post-stroke<br>• Fazekas scale | NIHSS at baseline and MoCA, WAIS-III, WMS-III, Semantic Association of Verbal Fluency Test, and the WCST at 3 months and 1 year | • A higher mCSVD score independently predicted classification into the low cognitive performance group |

**Table 3.** *Cont.*

| First Author (Year) | Type of Stroke, Study Design, Follow-Up Time, Participants (n) | Patients' Demographics: Age (Years), Gender (M/F), Education (Years/Level), Marital/Occupational Status, Income, BMI | Time of MRI Acquisition/Leukoaraiosis or WHM Assessment | Clinical and/or Psychometric Scales | Main Results |
|---|---|---|---|---|---|
| 6. Appleton (2020) [54] | • Any type (ischemic, hemorrhagic) <br>• Longitudinal <br>• 90 days <br>• 4011 patients | • Age: 70.3 ± 12.2 <br>• 2297 M/1714 F <br>• Education: n/a | • At baseline, usually before randomization <br>• Likert-scale for lucency (0 = no lucency, 1 = lucency restricted to region-adjoining ventricles, or 2 = lucency covering entire region from lateral ventricle to cortex) | NIHSS at baseline, mRS at day 90, t-MMSE, TICS-M, and verbal fluency (animal naming) | • LA score was independently associated with worse cognitive scores for t-MMSE and TICS-M at 90 days and for all SVD markers. <br>• Only severe LA was associated with all three cognitive measures (t-MMSE, TICS-M, verbal fluency) |
| 7. Suda (2020) [55] | • AIS (NIHSS ≤ 3) <br>• Longitudinal <br>• Until discharge <br>• 112 patients | • Age: 70 (IQR 61–79) <br>• 75 M/37 F <br>• Education: 12 (IQR 12–16) | • On admission <br>• Fazekas scale | NIHSS at admission, MoCA within 5 days of onset, mRS at discharge, and IQCODE pre-stroke | • Periventricular WMHs were significantly higher or more frequent in patients with cognitive impairment than in those without <br>• There were no significant differences between the cognitive impairment and non-cognitive impairment groups for the other SVD markers |

**Table 3.** *Cont.*

| | First Author (Year) | Type of Stroke, Study Design, Follow-Up Time, Participants (n) | Patients' Demographics: Age (Years), Gender (M/F), Education (Years/Level), Marital/Occupational Status, Income, BMI | Time of MRI Acquisition/Leukoaraiosis or WHM Assessment | Clinical and/or Psychometric Scales | Main Results |
|---|---|---|---|---|---|---|
| 8. | Yatawara (2020) [56] * | • AIS (mRS $\leq$ 2 at time of discharge)<br>• Longitudinal<br>• 6 months<br>• 185 patients | • Age: 57.6 $\pm$ 11.44<br>• 128 M/57 F<br>• Education: 9.30 $\pm$ 3.29 | • At initial presentation<br>• Fazekas scale | mRS, IQCODE, and MoCA | • WMHs were not associated with PSDem after controlling for demographics, cardiovascular risk, and global cortical atrophy.<br>• The risk of PSDem in patients with large infarcts, particularly in the subcortical regions, was 5.85 times higher when chronic confluent periventricular WMHs were present |
| 9. | Du (2019) [57] | • Any type<br>• Cross-sectional<br>• 127 patients | • Age: MCI group—65.34 $\pm$ 7.08, NCI group—65.29 $\pm$ 7.32<br>• 98 M/29 F<br>• Education: MCI group—10.12 $\pm$ 2.72, NCI group—11.31 $\pm$ 2.97 | • Within 1 week post-stroke<br>• Fazekas scale | MMSE, MoCA, TMT A and B, Stroop Color and Word Test, ROCFT, Boston Naming Test, and HAM-D | • Patients with mild cognitive impairment had significantly higher WMH Fazekas scores |
| 10. | Molad (2019) [58] | • AIS<br>• Longitudinal<br>• 2 years<br>• 397 patients | • Age: 66.9 $\pm$ 9.7<br>• 226 M/171 F,<br>• Education: cognitively intact group—13.6, PSCI group—11.3<br>• BMI: cognitively intact—26.9/PSCI—27.3 | • Within 7 days post-stroke<br>• Volumetric, Fazekas scale | NIHSS, MoCA, and NeuroTrax computerized cognitive testing | • WMH volume was among the predictors of post-stroke cognitive impairment and it remained a strong predictor following adjustment for age, gender, education, and NIHSS at admission |

**Table 3.** *Cont.*

| First Author (Year) | Type of Stroke, Study Design, Follow-Up Time, Participants (n) | Patients' Demographics: Age (Years), Gender (M/F), Education (Years/Level), Marital/Occupational Status, Income, BMI | Time of MRI Acquisition/Leukoaraiosis or WHM Assessment | Clinical and/or Psychometric Scales | Main Results |
|---|---|---|---|---|---|
| 11. Liang (2019) [59] | • AIS<br>• Longitudinal<br>• 15 months<br>• 451 patients | • Age: 66 ± 10.3<br>• 252 M/199 F<br>• Education: 6 (IQR 4–9) | • Within 7 days of admission<br>• Fazekas scale | MMSE at 3, 9, and 15 months | • The SVD burden was associated with MMSE scores and cognitive impairment. Among the SVD markers, WMHs were the most robust predictor of decreases in MMSE scores and cognitive impairment |
| 12. Zamboni (2019) [60] | • TIA or minor stroke (NIHSS < 4)<br>• Longitudinal<br>• 60 months<br>• 566 patients | • Age: 66.7 (20–102)<br>• 286 M/280 F<br>• Education: 13.0 ± 3.5 | • n/a<br>• Volumetric, Fazekas scale | NIHSS at baseline and MoCA at 1, 3, 6, 12, 24, and 60 months | • WMH volumes were strongly associated with cognitive status in patients aged ≤80 years but not in patients aged >80 years |
| 13. Hawe (2018) [61] | • Any type (ischemic n = 78, hemorrhagic n = 4)<br>• Longitudinal<br>• 6 months<br>• 82 patients | • Age: 60.6 ± 13.1<br>• 59 M/23 F<br>• Education: n/a | • n/a<br>• Volumetric | Chedoke–McMaster Stroke Assessment, Thumb Localizing Test, Behavioral Inattention Test, MoCA, and FIM | • The impact of WMHs was specific to cognitive impairments. Apart from the robotic position sense task, neither lesion volume nor WMH measures showed a significant ability to predict outcomes at 6 months compared to using initial impairment as measured by robotic assessments alone |
| 14. Puy (2018) [62] | • Any type (acute cerebral infarct or hemorrhage)<br>• Longitudinal<br>• 6 months<br>• 356 patients | • Age: 63.6 ± 10.6<br>• 216 M/140 F,<br>• Education: 10.8 ± 2.73<br>• BMI: 27.5 ± 4.6 | • At 6 months<br>• Fazekas scale | NIHSS, mRS, MMSE, MoCA, and optimized GCS | • WMH burden was among the factors related to the GCS, but the WMH burden and the presence of microbleeds were not independent role determinants |

**Table 3.** *Cont.*

| | First Author (Year) | Type of Stroke, Study Design, Follow-Up Time, Participants (n) | Patients' Demographics: Age (Years), Gender (M/F), Education (Years/Level), Marital/Occupational Status, Income, BMI | Time of MRI Acquisition/Leukoaraiosis or WHM Assessment | Clinical and/or Psychometric Scales | Main Results |
|---|---|---|---|---|---|---|
| 15. | Yatawara (2018) [63] * | • AIS<br>• Longitudinal<br>• 6 months<br>• 150 patients | • Age: PSDem group—62.11 ± 9.63, non-PSDem group—58.87 ± 8.43<br>• 105 M/45 F<br>• Education: PSDem group—7.77, non-PSDem group—9.77 | • At initial clinical presentation<br>• Fazekas scale | IQCODE, PHQ-9, and mRS | • Deep WMHs were indirectly associated with PSDem through disruption of executive functions, memory, and language.<br>• Periventricular WMHs were directly associated with PSDem and not mediated by deficits in cognitive domains |
| 16. | Divya (2017) [64] | • TIA or AIS (NIHSS ≤ 5)<br>• Longitudinal<br>• 3 months<br>• 50 patients/ 27 controls | • Age: 65.0 ± 9.3<br>• 41 M/9 F<br>• Education: 10.4 ± 3.4 | • On admission<br>• Fazekas scale | NIHSS, mRS, Malayalam version of Addenbrooke's Cognitive Examination, WMS verbal and visual subsets, RAVLT, Delayed Matching to Sample Task 48, attention span, TMT A and B, WCST, HADS, and Scale for the Instrumental Activities of Daily Living | • There were significant differences in the verbal list learning, visual learning, and paragraph recall scores between subjects with mild WMHs and with moderate-to-severe WMHs<br>• No significant differences were found with the presence or absence of WMHs and cerebral microbleeds for the selected neuropsychological test variables |
| 17. | Cao (2017) [65] | • Lacunar infarct<br>• Longitudinal<br>• 1 week after MRI<br>• 55 patients | • Age: 67.49<br>• 40 M/15 F<br>• Education: 10.40 ± 3.23 | • Average time: 105.6 ± 10.7 days<br>• Fazekas scale | NIHSS, HDRS, Trail-Making Test, Stroop Color and Word Test, category verbal fluency test, RAVLT, BNT, and ROCFT | • Periventricular diffusivity appeared to be an independent predictor of memory and visual–spatial processing, albeit to a lesser degree |

**Table 3.** *Cont.*

| First Author (Year) | Type of Stroke, Study Design, Follow-Up Time, Participants (n) | Patients' Demographics: Age (Years), Gender (M/F), Education (Years/Level), Marital/Occupational Status, Income, BMI | Time of MRI Acquisition/Leukoaraiosis or WHM Assessment | Clinical and/or Psychometric Scales | Main Results |
|---|---|---|---|---|---|
| 18. Molad (2017) [66] | • TIA or AIS (NIHSS < 17)<br>• Longitudinal<br>• 1 year<br>• 266 patients | • Age: 66.4 ± 9.4<br>• 162 M/104 F,<br>• Education: 13.6 ± 3.8<br>• BMI > 25: n = 178 | • Within 7 days<br>• Fazekas scale | NIHSS and NeuroTraxTM computerized cognitive testing | • WMH score showed significant negative associations with all cognitive domains except for verbal function, and the associations remained significant following adjustment for age, education, NIHSS during admission, gender, history of hypertension, and evaluation of cerebral atrophy using normalized ventricular CSF volume<br>• Adding other SVD markers to WMH did not considerably improve the prediction model |
| 19. Sivakumar (2017) [67] | • TIA or AIS (NIHSS ≤ 3)<br>• Longitudinal<br>• 90 days<br>• 115 patients | • Age: 66<br>• 75 M/40 F<br>• Education: n/a | • Within 72 h and at day 7 and day 30<br>• Volumetric, Fazekas scale | NIHSS, MoCA, mRS, and GDS | • WMH volumes predicted MoCA scores at day 30 and day 90<br>• WMH volumes predicted persisting cognitive impairment at day 30 and day 90, but when adjusted for age, WMH volumes no longer predicted cognitive impairment at day 30 |

**Table 3.** *Cont.*

| First Author (Year) | Type of Stroke, Study Design, Follow-Up Time, Participants (n) | Patients' Demographics: Age (Years), Gender (M/F), Education (Years/Level), Marital/Occupational Status, Income, BMI | Time of MRI Acquisition/Leukoaraiosis or WHM Assessment | Clinical and/or Psychometric Scales | Main Results |
|---|---|---|---|---|---|
| 20.   Zhang (2017) [68] | • AIS (NIHSS ≤ 5)<br>• Longitudinal<br>• 30 days<br>• 217 patients | • Age: none-to-mild LA—59.8 ± 4.2, severe LA—67.5 ± 15.5<br>• 147 M/70 F<br>• Education: none-to-mild LA vs. severe LA: illiterate—30% vs. 31%, primary school—26.7% vs. 29%, middle school or higher—43.3% vs. 40% | • Within 48 h of admission<br>• Fazekas scale | NIHSS and MMSE at baseline and at 30 days | • LA burden, irrespective of chronological age or infarct volume or location, was associated with worse functional and cognitive recovery after an initial minor ischemic stroke since the MMSE score improved in the none-to-mild group but not in the severe group at day 30<br>• Cognitive impairment on admission was more prevalent in the none-to-mild LA group than in the severe LA group, while at day 30, this was reversed |
| 21.   Mandzia (2016) [69] | • TIA or minor stroke (NIHSS < 4)<br>• Longitudinal<br>• 90 days<br>• 92 patients | • Age: 65.1 ± 12.03<br>• 68 M/24 F<br>• Education: 14 (IQR 12–16) | • At baseline<br>• Volumetric | mRS and CES-D | • WMH volume was not associated with 90-day cognitive performance |

**Table 3.** *Cont.*

| | First Author (Year) | Type of Stroke, Study Design, Follow-Up Time, Participants (n) | Patients' Demographics: Age (Years), Gender (M/F), Education (Years/Level), Marital/Occupational Status, Income, BMI | Time of MRI Acquisition/Leukoaraiosis or WHM Assessment | Clinical and/or Psychometric Scales | Main Results |
|---|---|---|---|---|---|---|
| 22. | Moulin (2016) [70] ** | • ICH<br>• Longitudinal<br>• Median of 6 years<br>• 218 patients | • Age: 67.5 (IQR 55–76)<br>• 118 M/100 F<br>• Education (≤8 y): dementia-free group—n = 65, group that developed dementia—n = 22 | • Soon after admission<br>• Fazekas scale | IQCODE, mRS, NIHSS, and MMSE | • Severe LA was an independent risk factor for new-onset dementia after intracerebral hemorrhage |
| 23. | Benedictus (2015) [71] ** | • ICH<br>• Longitudinal<br>• Median of 4 years<br>• 167 patients | • Age: 64 (IQR 53–75)<br>• 98 M/69 F<br>• Education ≤8years: n = 92 | • Soon after admission<br>• Fazekas scale | IQCODE, mRS, MADRS, and MMSE | • Factors associated with cognitive decline in univariate analyses were previous stroke or TIA, preexisting cognitive impairment, presence of microbleeds, severity of WMHs, and severity of cortical atrophy<br>• In multivariable analyses, only previous stroke or TIA, preexisting cognitive impairment, and severity of cortical atrophy remained independent prognostic factors |

**Table 3.** *Cont.*

| | First Author (Year) | Type of Stroke, Study Design, Follow-Up Time, Participants (n) | Patients' Demographics: Age (Years), Gender (M/F), Education (Years/Level), Marital/Occupational Status, Income, BMI | Time of MRI Acquisition/Leukoaraiosis or WHM Assessment | Clinical and/or Psychometric Scales | Main Results |
|---|---|---|---|---|---|---|
| 24. | Kumral (2015) [72] | • Any type (large-artery disease, small-artery disease, cardioembolism, intracerebral hemorrhage, other stroke types)<br>• Longitudinal<br>• 5 years<br>• 8784 patients | • Age: 66.2 ± 12.6<br>• 5387 M/3397 F<br>• Education: n/a | • n/a<br>• Fazekas scale | NIHSS at admission | • All patients with LA had an increased probability of mild cognitive impairment and dementia compared to those without LA<br>• Probability of cognitive decline was significantly higher in patients with severe LA compared to those with mild/moderate LA |
| 25. | Nakano (2015) [73] | • Any type (ischemic n = 107, ICH n = 5)<br>• Longitudinal<br>• 4.8 years<br>• 112 patients | • Age: 73.6 ± 10.4<br>• 69 M/43 F,<br>• Education: 12.5 ± 2.4<br>• BMI: 23.6 ± 3.2 | • n/a<br>• Fazekas scale | MMSE, Revised Hasegawa Dementia Rating Scale, FAB, GDS, and apathy scale | • The severity of WMHs was significantly higher among converters to post-stroke dementia |

Notes. Studies from the same group or cohort are highlighted with * and **. n/a = not available; n = number; M/F = male/female; MRI = magnetic resonance imaging; WHMs = white matter hyperintensities; TIA = transient ischemic attack; NIHSS = National Institutes of Health Stroke Scale; MoCA = Montreal Cognitive Assessment; PSD = post-stroke depression; IQCODE = Informant Questionnaire on Cognitive Decline in the Elderly; PHQ-9 = Patient Health Questionnaire-9; mRS = Modified Rankin Scale; AIS = acute ischemic stroke; IQR = interquartile range; MMSE = Mini-Mental State Examination; SVD = small vessel disease; LA = leukoaraiosis; FIM = Functional Independence Measure; NfL = neurofilament light chain; MoCA = Montreal Cognitive Assessment; ICH = intracerebral hemorrhage; FAB = Frontal Assessment Battery; GDS = Geriatric Depression Scale; CES-D = Center for Epidemiologic Studies Depression; t-MMSE = Telephone MMSE; TICS-M = Modified Telephone Interview for Cognitive Status; CSF = cerebrospinal fluid; RAVLT = Rey Auditory Verbal Learning Test; TMT A and B = Trail-Making Test parts A and B; WCST = Wisconsin Card Sorting Test; HADS = Hospital Anxiety Depression Scale; MADRS = Montgomery–Åsberg Depression Rating Scale; PSCI = post-stroke cognitive impairment; cSVD = cerebral small vessel disease; WAIS = Wechlser Adult Intelligence Scale; WMS = Wechsler Memory Scale; ROCFT = Rey–Osterrieth Complex Figure Test; MCI = mild cognitive impairment; NCI = no cognitive impairment; HDRS = Hamilton Depression Rating Scale; BNT = Boston Naming Test; mNOS = Modified Newcastle–Ottawa Scale.

**Table 4.** Characteristics of included studies focusing on depression and cognition.

| First Author (Year) | Type of Stroke, Study Design, Follow-Up Time, Participants (n) | Patients' Demographics: Age (Years), Gender (M/F), Education (Years/Level), Marital/Occupational Status, Income, BMI | Time of MRI Acquisition/Leukoaraiosis or WHM Assessment | Clinical and/or Psychometric Scales | Main Results |
|---|---|---|---|---|---|
| 1. Douven (2018) [74] | • Any type (non-fatal ischemic or hemorrhagic stroke) <br> • Longitudinal <br> • 1 year <br> • 245 patients | • Age: stroke only group—65.9 ± 11.0, executive dysfunction only group—69.9 ± 11.4, PSD group—64.7 ± 11.3, depression–executive dysfunction syndrome group—67.7 ± 13.5 <br> • 157 M/88 F <br> • Education: low—n = 101, middle—n = 86, high—n = 58 | • At baseline <br> • Volumetric | MMSE | • Patients with "depression–executive dysfunction syndrome" showed higher WMH volumes compared to all other groups |

Notes. n/a = not available; n = number; M/F = male/female; MRI = magnetic resonance imaging; WHMs = white matter hyperintensities; PSD = post-stroke depression; MMSE = Mini-Mental State Examination; mNOS = Modified Newcastle–Ottawa Scale.

Regarding the method of LA neuroimaging assessment, 27 studies preferred the Fazekas score, 1 used the Age-Related White Matter Change Scale (ARWMCS), 1 used the Wahlund scale score, 1 described a new scoring system, and 10 estimated the volume of WMHs using MRI.

### 3.3. Study Design

Almost all the studies included in this review were longitudinal, and they employed either retrospective or prospective groups. Only one study was cross-sectional.

### 3.4. Stroke Patient Groups and Demographic Profiles

The total numbers of stroke patients included in the studies ranged from n = 55 to n = 8784. Among the 34 studies, 7 had disease sample sizes between 1 and 100 patients, 10 between 101 and 200, 5 between 201 and 300, 4 between 301 and 400, and 8 had disease sample sizes larger than 400 patients. Mean/median patient ages ranged from 46 years to 73.6 years.

### 3.5. Reference Groups

Among the 34 studies, stroke patients were contrasted with demographically matched healthy individuals in only 2 studies, with the rest of them (25/27 studies) not including a healthy control group. None of the studies included a disease-control group other than stroke patients.

### 3.6. Time of MRI Execution

In 11 studies, the MRI test was performed at baseline. It was performed in the first 48 h in one study, within the first 5 days following stroke onset in one study, within the first week in seven studies, in the first month in two studies, within 3 months from the onset of symptoms in two studies, at least 3 months after stroke in one study (with an average time of 105.6 ± 10.7 days), and at 6 months in one study. MRI was executed at baseline and then repeated at 15 months in one study and it was performed within the first 72 h following stroke onset and then repeated at day 7 and at day 30 in one study.

## 4. Discussion

A literature review of studies published over the last decade was conducted to elucidate the prognostic value of LA for treating depression and cognitive impairment in stroke settings (Figure 2). A total of 34 original full-text articles dealing with the potential utility of the assessment of pre-existing LA in relation to the emergence of depression and cognitive impairment in stroke patients were identified and divided into two groups.

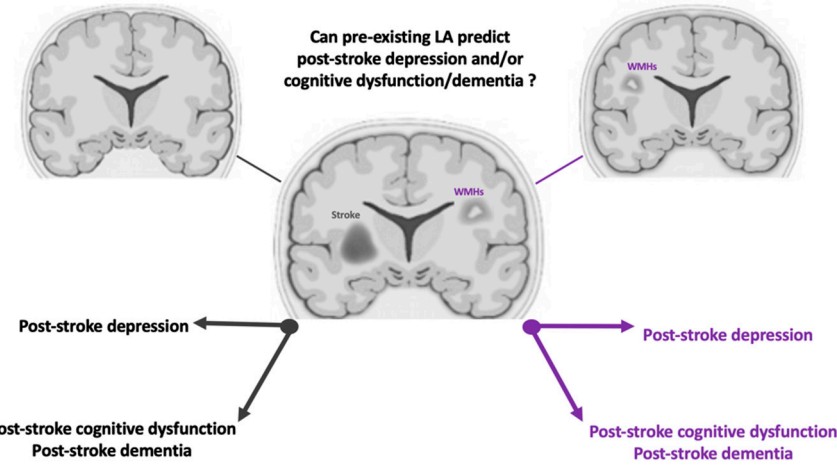

**Figure 2.** Graphical representation/summary of the aim of the review.

*4.1. Depression and White Matter Hyperintensities*

With respect to post-stroke outcomes, Douven and colleagues [74] examined whether the co-occurrence of executive dysfunction (DES) and PSD showed different associations with neuroimaging markers, the course of depression, and executive function. They also questioned whether this co-occurrence was associated with different courses in other cognitive domains and with quality of life. An analysis of one-year follow-up data showed that depression–executive dysfunction syndrome was associated with left-sided lesions, larger WMH volume, greater global brain atrophy, increased frequency of old infarcts, and significant burden from cerebral small vessel disease (cSVD) [74]. According to the authors, this suggested that DES is associated with increased generalized (vascular) brain pathology. Furthermore, patients with depressive–executive dysfunction syndrome showed more functional impairments at baseline compared to patients with PSD only and had a chronic course of depressive symptoms. As with the executive dysfunction-only group, patients with depressive–executive dysfunction syndrome also had poor cognitive performance that did not improve over time [74]. Depressive symptoms only improved in the PSD-only group. This was in agreement with findings from late-life depression patients in whom executive dysfunction was associated with relapse and recurrence of depression [75]. Based on these findings, the authors noted that it might be useful to screen for the presence of both executive dysfunction and PSD in clinical practice, since this might identify stroke survivors who need special attention.

In another study, Douven and colleagues [45] explored imaging markers of lesion-related and generalized brain pathology over a one-year period, along with the development of post-stroke apathy and PSD. Stroke patients with high levels of global brain atrophy or cSVD burden at baseline had elevated odds of exhibiting post-stroke apathy, regardless of stroke lesion volume or co-morbid PSD. Additionally, patients with greater global brain atrophy or medium cSVD burden showed elevated odds of developing PSD after adjusting for relevant confounders. After accounting for co-morbid post-stroke apathy and stroke volume, only medium cSVD burden remained significantly associated with PSD [45]. Furthermore, there was no association between PSD/post-stroke apathy and lesion-related imaging markers (i.e., location, type, volume). Thus, global brain atrophy and total cSVD burden are significant predictors of post-stroke apathy [45].

Tanislav and colleagues [49] investigated the depressive symptomatology and its determinants in young stroke patients at the acute stage. WMHs were not found to be associated with the occurrence of clinically related depressive symptoms [49], which was in line with previous findings that, even though WMHs and post-stroke delayed mood disorders are associated, the development of depressive symptoms in the acute stage is not definitely related to WMHs [76]. Moreover, acute ischemic lesions were not found to be associated with clinically related depressive symptoms, in contrast to previous findings [76,77]. Moreover, female patients suffered more often from clinically related depressive symptoms than male patients. Similar findings have been reported in previous studies conducted in general stroke populations [78–80] where many stroke patients were older women living alone [78,80]. Furthermore, in light of the psychological impact a cerebrovascular event has on patients, gender-specific coping strategies might serve as explanations for the differences in clinically related depressive symptoms between male and female patients [49]. In addition, a tendency for older patients to develop depressive symptoms after stroke more often has been observed, in line with previous studies [80]. Notably, Tanislav and colleagues [49] reported that most patients with clinically related depressive symptoms were found to be in the oldest age group of 45–55, regardless of gender. Overall, these outcomes suggest that, in young stroke patients, clinically related depressive symptoms at the acute stage are related to a reactive psychological phenomenon in response to the event rather than to pre-existing or acute brain lesions.

A recent study by Bae and colleagues [46] examined the effect of periventricular and deep WMHs on depressive symptoms in the acute and chronic phases of stroke; i.e., 2 weeks and 1 year post-stroke. Periventricular WMHs were significantly associated with PSD at

2 weeks (acute phase), whereas deep WMHs were related to PSD 1 year post-stroke (chronic phase) [46]. Previous studies have shown that the risk factors for PSD differ according to the time after the stroke [81]. The current findings highlight the dynamic nature of PSD and the need for continuous monitoring of stroke patients' mood state [82]. Finally, in terms of sex, females alone were associated with major PSD at follow-up.

Moreover, a recent study by Jaroonpipatkul and colleagues [43] examined whether post-stroke depressive symptoms measured 3 months post-stroke using the Montgomery–Åsberg Depression Rating Scale (MADRS) were influenced by WMHs measured by fluid-attenuated inversion recovery (FLAIR) and infarction volume measured by diffusion-weighted imaging (DWI). Despite only mild elevations in MADRS scores 3 months after the acute infarction, stroke patients had significantly higher scores than controls. Additionally, total WMHs showed no significant direct effect on the depressive symptoms measured 3 months post-stroke. However, there was a significant indirect effect of WMHs on key depressive symptoms, such as concentration–tension symptoms and lassitude. Bilateral DWI stroke volumes and their accompanying disabilities led to these latter effects. Moreover, age, ischemic heart disease, and hypertension explained 37.5% of the variance in total WMHs, and these premorbid features also explained 30% of the variance in total DWI stroke volume [43]. Thus, age and hypertension are strongly linked to critical depressive symptoms (i.e., concentration–tension symptoms and lassitude), which are also related to higher WMH volume, greater stroke infarct volume, and the consequent disabilities.

Focusing on lacunar infarcts, WM lesions, cerebral microbleeds, and enlarged perivascular spaces, Guo and colleagues [42] investigated the relationship between thyroid function profiles and PSD and evaluated whether cSVD mediates the relationship between thyroid function profiles and PSD in patients with acute ischemic lacunar stroke. Serum TSH levels on admission were likely to predict depression after acute ischemic lacunar stroke, whereas cSVD mediated the association between TSH and PSD [42]. Therefore, combining TSH and cSVD may be a reasonable and helpful approach for the future assessment and prevention of PSD [42]. Acknowledging the risk factors associated with PSD would help in identifying the disease and designing specific interventions in advance.

Furthermore, Zhou and colleagues [44] assessed the effects of total cSVD on early-onset depression after an AIS and developed a new nomogram including total cSVD burden to predict early-onset PSD [44]. They found that WMHs, silent lacunar infarction, cerebral microbleeds, and enlarged perivascular spaces were all related to early-onset PSD. Overall, several mechanisms may explain the association between cSVD and PSD. For instance, in patients with pre-stroke cSVD, hypertension and diabetes are more prevalent, and they are closely associated with brain dysfunction [83].

Regarding TIA, Carnes-Vendrell and colleagues [47] examined the prevalence, development, and predictors of PSD and post-stroke apathy over a 12 month follow-up period. This research study was one of the few longitudinal studies focused on depression and apathy after TIA and minor stroke. Although TIA patients do not have persistent functional deficits after stroke, they may suffer from neuropsychiatric complaints, such as post-stroke apathy and PSD [47]. Depression levels appeared to be relatively high at the acute stage after TIA and minor stroke but generally decreased during the follow-up period compared to apathy. Hyperintensities were only noticeable for deep WM scores in the bivariate analysis of PSD, which was conducted after 12 months. There were no associations at baseline between any other neuroimaging variables and the development of PSD.

The findings of Pavlovic and colleagues [48] are worthy of mention as the researchers sought to investigate baseline predictors of depression after long-term follow-up in patients with SVD who initially presented with their first-ever lacunar stroke and were not depressed or cognitively impaired at that time. Data showed that patients with depression were more frequently diagnosed with cognitive decline than non-depressed ones. In addition, all MRI parameters regarding WMH severity and the total number of lacunar infarcts were higher in SVD patients who were depressed than non-depressed patients. Furthermore, these findings confirmed that depression can be predicted based on baseline

WMH burden and the number of lacunar infarcts detected by MRI, with WMHs remaining an independent predictor.

Additionally, a 3 year follow-up study showed a significant association between the progression of WMHs and incident depression, supporting the vascular hypothesis and suggesting that WMHs have a causal role in late-onset depression [37]. A gender-specific influence was not detected regarding depression occurrence, which was in line with the work of Forlani and colleagues [84], who also did not notice gender differences in the overall prevalence of late-life depression [84]. Nonetheless, the associations between depression and functional measures were more robust in male than female patients [84]. Ultimately, as highlighted by a recent systematic review by Kutlubaev and Hackett [85], no consistent correlation between depression after stroke and gender has been documented.

### 4.2. Cognitive Impairments and White Matter Hyperintensities

In addition to the possible onset of depressive symptoms after a stroke, the onset of cognitive impairments and further decline is also very often encountered. Markers such as LA can help to prevent and identify the signs of post-stroke cognitive impairment. For instance, Kumral and colleagues [72] examined whether LA contributes to the occurrence of a specific type of cognitive disorder after an initial stroke. The primary finding was that a substantial proportion of patients with LA suffered from notable cognitive disorders, such as mild cognitive impairment (MCI) or vascular and mixed dementia, within five years of their initial stroke as opposed to those who did not have LA. Moreover, in addition to other risk factors for patients with MCI and dementia, age was associated with decreased cognitive functioning. According to multiple previous studies, age-associated LA strongly correlates with cognitive decline, progression from MCI to dementia, a poor functional outcome following a stroke, and the risk of recurrent hemorrhage in patients with stroke [35, 86,87]. LA was also independently associated with poorer cognitive outcomes at five years and, after the initial stroke, the rate of LA-related dementia increased during the 5 year time period [72]. Notably, these findings highlight that there is a high likelihood for stroke among the elderly to result in long-term disability and handicaps with impacts on health and social care services in the future [72].

Furthermore, Hawe and colleagues [61] used robotics and clinical measurements to examine the effects of acute lesion volume and WMH volume on longitudinal recoveries post-stroke [61]. Interestingly, acute lesion volume and WMH volume influenced different domains of recovery post-stroke. Overall, acute lesion volume affected sensory and motor impairments post-stroke, as well as general function. On the other hand, the effect of WMH volume was limited to cognitive deficits. Furthermore, although lesion volume and WMH volume negatively impacted recovery over time, they provided no valuable information for the prediction of outcomes 6 months after stroke, suggesting that they might have limited prognostic value [61]. However, the findings provided evidence that WMH volume affects cognitive aspects of recovery to a greater extent than lesion volume.

Puy and colleagues [62] examined the neuroimaging determinants of post-stroke global cognitive performance and the relative contributions of a spectrum of MRI markers, such as lesion burden and strategic location. Among the many different MRI markers, specific lesions within strategically located brain regions (e.g., right corticospinal tract, left antero-middle thalamus, left arcuate fasciculus) were most predictive of post-stroke cognitive function and predicted 22.5% of the variance in the global cognitive score [62]. In addition, stroke volume, total medial temporal lobe atrophy, and brain tissue volume were independent but weaker predictors. Finally, the WMH burden and the presence of microbleeds were not independent determinants. These two factors were found to have weak associations with cognitive performance and were only statistically significant in the univariate analyses. Furthermore, there was a lack of evidence regarding a threshold effect for the WMH burden or the number of microbleeds. Thus, these findings suggest that WMHs and brain microbleeds make only minor contributions to post-stroke neurocognitive

disorders in the stroke population, and it is likely that stroke lesions might override the effect of WMHs [62].

Additionally, Appleton and colleagues [54] studied a large group of patients with lacunar stroke syndromes and found that baseline imaging markers of SVD and neurocognitive decline were common, individually and collectively, and negatively associated with poor functional and cognitive outcomes at 90 days. Furthermore, as the specificity of the diagnosis of lacunar stroke increased, the association between SVD scores and inadequate functional outcomes was augmented, and brain fragility showed similar associations across the population. Finally, their data confirmed the critical prognostic value of the three brain frailty measures used in CT; i.e., LA, atrophy, and old vascular lesions were independently associated with poor outcomes in IST-3. Moreover, these imaging markers were also associated with functional and cognitive outcomes 90 days after stroke when they were pooled as a score that included old infarcts [54]. Considering that these imaging markers can be easily detected using CT by physicians and radiologists in the treatment of acute stroke and that they may have solid prognostic significance, they may prove helpful for predicting cognitive and functional outcomes when added to other clinical markers.

In a recent, similar study, Georgakis and colleagues [88] aimed to determine whether the global burden of SVD assessed from baseline MRI could predict cognitive and functional outcomes up to 12 months after a stroke. The presence and severity of SVD burden in baseline MRI were associated with poor post-stroke cognitive and functional outcomes [88]. Specifically, the global SVD burden score and individual SVD markers were associated with cognitive and functional impairments up to 12 months after a stroke. In addition, patients with a higher SVD burden at baseline performed worse in tests of executive function, attention, language, and visuospatial ability within the same time period. Furthermore, SVD was found to be an independent risk factor for post-stroke outcomes. Individual lesions contributed independently to poor outcomes, and there was a dose–response relationship for all lesion types, with the most substantial relationship found for lacune count [88]. A more profound exploration revealed that a model considering the severity rather than the presence of individual SVD lesions improved the prediction ratio. This finding has implications for future research, emphasizing the need to design and develop more efficient tools for SVD burden quantification [88].

Meaningful findings emerged from the study by Du and colleagues [57], who examined the mechanisms underlying early cognitive impairment in a post-stroke non-dementia cSVD cohort by comparing the SVD score with measures of structural brain networks. Using detailed neuropsychological and multimodal MRI measures, it was noted that the correlation between SVD score and cognitive functions was not significant. Moreover, regarding cerebral vascular brain injuries, only lacunes and WMHs are consistently associated with cognitive deficits [89]. As shown in Du et al.'s study, the latter factor was significantly related to multiple cognitive domains [57]. Overall, the researchers' efforts provided some evidence for the mechanisms underlying early cognitive impairment in SVD. Namely, it was found that brain network measures were independent surrogate markers for cognitive function, acting as mediators between conventional CVBIs. In conclusion, the authors strongly argue that, to understand SVD-related cognitive impairments, it may be more insightful to study brain networks rather than individual lesions [57].

Regarding dementia, Nakano and colleagues [73] investigated the annual ratio of conversion into PSDem among cognitively normal patients at baseline and compared the risk factors for conversion or reversion. During the follow-up period, a subgroup of participants developed PSDem and was labeled "converters". The remaining participants had normal cognitive functions and were characterized as "non-converters". MRI findings highlighted that the severity of WMHs was significantly higher in the converters than non-converters [73]. Additionally, when comparing non-converters to converters, it became evident that the latter group were older, had worse baseline scores in the Mini-Mental State Examination (MMSE) and Revised Hasegawa Dementia Rating Scale (HDS-R), low BMIs, and more severe WMHs [73]. The aforementioned characteristics were considered risk

factors for PSDem. Recent studies have highlighted other risk factors for cognitive decline in post-stroke patients, such as education, occupation, recurrent stroke, PSD, post-stroke apathy, APOE ε4 status, infarcted volume in strategic areas (e.g., cortical limbic areas and heteromodal associative areas, including the frontal cortex and WM), and the presence of multiple lesions (e.g., [90–94]). However, in the study by Nakano et al., such factors were not significantly related to conversion.

Investigating the effects of cerebral microbleeds, diffusion tensor imaging (DTI), and brain volumetric measurements on cognitive functions in a cohort of post-stroke non-dementia SVD patients, Cao and colleagues [65] found that multiple domains of cognition, in addition to language, were affected in the early stages of vascular cognitive impairment (VCI); primarily, attention and executive functions. Additionally, hypertension and depression were detected more often in cognitively impaired patients. In this study, one independent predictor of executive function was periventricular WM diffusivity, a more sensitive index than deep cerebral microbleed counts. Furthermore, periventricular diffusivity was an independent predictor of memory and visual–spatial processing, albeit to a lesser degree. Thus, disruption of WM integrity independently affects executive and memory functions in SVD patients [65].

With regard to AIS, there is a plethora of articles presenting interesting and significant evidence concerning this type of stroke, WMHs, and cognitive dysfunctions. For instance, Liang and colleagues [59] sought to investigate the association between SVD burden, a combination of multiple SVD markers, and cognitive dysfunction after a stroke. SVD burden was found to be an alternative surrogate marker of cognitive dysfunction, at least in the first year after stroke, in patients with mild-to-moderate AIS. Thus, the SVD score was suitable for screening stroke survivors at risk of cognitive dysfunction or dementia. Additionally, WMHs showed the most robust relationship with cognitive dysfunction post-stroke among the four SVD markers examined. These findings reinforce previous research linking SVD burden with cognitive dysfunction after stroke and underline the potential of SVD markers for treatment in clinical trials to decrease the risk of cognitive decline after stroke [59].

Furthermore, Peng and colleagues [52] evaluated the predictive value of the combination of a serum biomarker for axonal damage (i.e., neurofilament light chain (NfL)) and neuroimaging markers (volume of infarction and WMHs) for neuronal abnormalities in post-stroke cognitive outcomes [52]. NfL serum levels measured within one month after a stroke could potentially function as a moderate predictor of cognitive outcomes upon discharge after acute stroke rehabilitation. In addition, a biomarker panel combining NfL serum levels and MRI markers (i.e., the volume of infarction and WMHs) was found to be able to improve the predictive value and facilitate precise ischemic stroke treatments for patients [52].

Another two studies confirmed the previous findings regarding cognitive impairments. Specifically, Suda and colleagues [55] evaluated the frequency of early cognitive impairment in patients with minor ischemic stroke, analyzed the factors associated with early cognitive impairment, and assessed functional outcomes. Along the lines of what emerged in the study by Bae and colleagues [46], periventricular WMHs had significantly higher volume or were more frequent in patients with cognitive impairment than in those without cognitive difficulties. Moreover, pre-existing temporal horn atrophy was independently associated with early cognitive impairment.

The second study offered meaningful data that confirmed the findings regarding LA and cognitive recovery after a stroke, as well as adding new information. Zhang and colleagues [68] studied 217 patients with AIS and found that LA burden, irrespective of chronological age or infarction volume or location, was associated with worse functional and cognitive recovery after an initial minor ischemic stroke. The degree of pre-existing LA modulated the association between infarction volume and neurological deficit severity as assessed with the National Institutes of Health Stroke Scale (NIHSS). Furthermore, cognitive impairment at admission was more prevalent in the none-to-mild LA group than

in the severe LA group, while, at 30 days, this trend was reversed [68]. According to one speculative explanation, patients with none-to-mild LA are more likely to have pre-existing cognitive impairments resulting from other factors that also increase their stroke risk. This could explain why they showed a higher prevalence of cognitive impairment at admission. It is also possible that psychological distress may interact with cognitive evaluation, and mood distress may lead to lower MMSE scores in patients who have better cognitive functions [95,96].

In a more recent study, Molad and colleagues [58] evaluated the contribution of vascular pathology to post-stroke cognitive impairment, separate from and in conjunction with AD-related measures. WMH volume was among the predictors of post-stroke cognitive impairment, and it remained a strong predictor following adjustment for age, gender, education, and NIHSS score at admission.

Finally, Sung and colleagues [53] examined the potential factors associated with post-stroke cognitive performance and trajectories. Patients with mild ischemic strokes occurring for the first time showed signs of cognitive improvement after a year. During follow-up, the generalizing estimating equation statistical model indicated that stroke severity, lesions involving the cortical region, stroke etiologies, higher cSVD burden (which represents the baseline total brain damage associated with stroke), and age-disproportional hippocampal atrophy did not significantly affect cognitive performance post-stroke. However, higher cSVD burden at baseline significantly predicted poor cognitive outcomes after stroke based on the cognitive trajectory model [53].

Returning to post-stroke dementia and AIS, observations and findings worth mentioning can be found in various studies. Yatawara and colleagues [63] examined how different patterns of association between anatomical lesions and cognitive domain impairments contribute to the risk of incident PSDem. There was a greater severity of pre-stroke lesions in PSDem participants, including WMHs and global cortical atrophy, compared to controls. Moreover, participants who developed PSDem exhibited a lower educational level, higher functional disability, and lower performance in global and domain-specific cognitive tests in comparison to controls. Deep WMHs were indirectly associated with incident PSDem as mediated by executive dysfunction and delayed memory deficits. Nevertheless, when controlling for executive functions and delayed memory performance, the direct relationship between deep WMHs and PSDem was attenuated, suggesting a complete mediation process [63]. Statistically, the odds ratio for incident PSDem among patients with deep WMHs and executive dysfunctions was 12 times greater than among patients without clinically relevant DWMHs or executive dysfunctions. Similarly, the OR for PSDem among patients with deep WMHs and memory deficits was three times greater than among patients without these conditions. Finally, periventricular WMHs were directly associated with PSDem but not mediated by any cognitive domain [63].

Interestingly, in a more recent study, Yatawara and colleagues [56] studied how different types of acute stroke-related infarcts interacted with specific chronic cerebrovascular pathologies in the development of early-onset PSDem. WMHs, chronic lacunes, microbleeds, and acute infarcts were not associated with PSDem after controlling for demographics, cardiovascular risk, and global cortical atrophy. Nonetheless, the risk of PSDem in patients with large infarcts, particularly in subcortical regions, was 5.85 times higher when chronic confluent periventricular WMHs were present. In accordance with the previously mentioned studies, patients who developed PSDem were older and had fewer years of education, greater global cortical atrophy scores, more confluent WMHs, and more lobar microbleeds than stroke patients who did not develop PSDem.

Other studies have focused on TIAs and minor strokes and explored the relationships between these conditions and WMHs and cognitive impairments. For example, previous studies have noted that MRI-detectable WMHs presumed to be of vascular origin are associated with cognitive impairments and dementia [38,97]. Based on these findings, Zamboni and colleagues [60] examined whether the MRI-detectable WMHs and cognitive status reported in the earlier studies persisted at older ages (>80 years) when there are

often reports of WM abnormalities in clinical practice. The associations between WMHs and cognitive status differed depending on the participants' age groups. A high WMH load was detected in patients with previous TIA or minor stroke aged over 80 years old. However, in this age group, this WMH burden was not significantly associated with cognitive impairment. High WMH load was strongly associated with cognition only among patients younger than 80 years old, who were four times more likely to have severe impairments than patients in the same age group with low WMH load [60]. Moreover, although MRI markers of WM damage have been used as a proxy of VCI [98,99], their interpretation becomes more complex in patients over 80.

Another interesting approach was described by Mandzia and colleagues [69], who examined clinical and imaging features associated with worse cognitive performance 90 days after a TIA or minor stroke. According to this study, cumulative cognitive scores at 90 days showed that patients' performances were within the normal range, and there was no difference in performance based on the clinical type of the event (i.e., clinical TIA versus stroke). However, there were several stroke- and TIA-related variables associated with worse cognitive performance in tests of executive function and psychomotor processing. There was no association between previous lacunar stroke and worse cognitive performance, possibly due to the limited number (19%) of patients who had subclinical lacunar infarcts predating their event. Additionally, WMH volume was not associated with worse cognitive performance. When compared to other studies, this cohort was younger and did not suffer from dementia, and a large proportion of the patients suffered from TIAs or minor cortical strokes and had smaller mean WMH volumes [69].

Another interesting issue is the combination of TIAs, minor strokes, and AIS in the same population. For instance, Sivakumar and colleagues [67] tested the hypothesis that persistent cognitive impairment after TIA/minor stroke can be predicted by the volume of diffusion-weighted imaging lesions. It is well-established that cognitive impairment is common after TIA/minor stroke. Many patients experience temporary deficits, while others continue to experience them 90 days later. In contrast to the primary hypothesis, chronic WM disease burden was more predictive of cognitive changes over time than acute ischemic DWI lesions. Specifically, the volume of WMHs predicted persistent cognitive deficits when patients were assessed more acutely. Moreover, there were no significant differences in acute ischemic lesion volumes between TIA/minor stroke patients with transient cognitive impairment or normal cognition and patients with persistent deficits. However, the former group of patients exhibited lower WMH volumes.

Other studies have provided further details regarding cognitive decline and WM lesions in patients with the abovementioned types of strokes [64,66]. Molad and colleagues [66] studied whether adding other SVD markers to WMHs would improve prediction models of post-stroke cognitive performances [66]. They found significant associations between SVD burden score, age, and Framingham Stroke Risk Profile scores, but not years of education. In addition, comparing male and female patients, they found higher SVD burden scores in the male group. No associations were found between cognition and lacune count, PVS score, or microbleed count. Moreover, all cognitive domains except verbal function were negatively associated with the WMH score based on the Fazekas scale. This association remained significant following adjustment for multiple factors. Overall, adding other SVD markers to WMHs does not improve the predictability of post-stroke cognition rates, and WMHs remain the only predictor when it comes to first-ever mild-to-moderate ischemic stroke or TIA.

Divya and colleagues [64] examined the cognitive profiles of post-stroke vascular MCI (vaMCI) patients compared to patients with MCI with a non-vascular etiology (non-vaMCI) and cognitively normal healthy controls [64]. The results that emerged were in agreement with the previously reported findings. There were significant differences in scores for the global cognitive test (Addenbrooke's Cognitive Examination (ACE)) and for verbal and visual memory measures and differences in numbers of omission errors in the Rey Auditory Verbal Learning Test between subjects with post-stroke vaMCI and the control

group. Moreover, subjects with mild WMHs exhibited significantly lower scores for verbal list learning, visual learning, and paragraph recall scores than those with moderate-to-severe WMHs. Finally, WMHs, cerebral microbleeds, and neuropsychological test variables were not significantly different among vaMCI and non-vaMCI patients [64].

Few studies have examined intracerebral hemorrhage (ICH) patients. Benedictus and colleagues [71], Moulin and colleagues [70], and Pasi and colleagues [51] have offered insightful information regarding cognitive impairments and brain lesions in patients who have survived ICH. The former study aimed to determine prognostic factors for cognitive decline in patients with ICH. Through the analysis of follow-up data, a strong association between pre-existing cognitive impairment, cortical atrophy, and stroke or TIA was detected, all of which are significant predictors of cognitive decline following an ICH; thus, prognostic factors may already exist prior to the occurrence of the ICH. Overall, 37% of the patients demonstrated cognitive decline during the follow-up. In univariate analyses, factors associated with cognitive decline were a previous stroke or TIA, pre-existing cognitive impairment, the presence of microbleeds, the severity of WMHs, and the severity of cortical atrophy [71]. Regarding patients admitted with an ICH, these findings emphasized the importance of carefully assessing pre-existing cognitive decline and cortical atrophy at admission. Future cognitive decline may influence clinical decisions affecting patients with these conditions concerning their long-term functional prognosis and treatment. Additionally, this group of patients may be a suitable study group for exploring the effectiveness of intensive preventive strategies for the attenuation of cognitive decline following an ICH [71].

Moulin and colleagues [70] examined the incidence of dementia and risk factors after an ICH. After following-up on the progress of the individuals for 6 years on average, it was found that the incidence of new-onset dementia reached 28.3% 4 years after the ICH. Furthermore, according to previous findings on pre-existing dementia [100], patients with lobar ICH (who had no signs of dementia before ICH) were more likely to develop new-onset dementia than those with non-lobar ICH. The results of this study indicate that there are existing risk factors for new-onset dementia before ICH, suggesting that new-onset dementia is not the result of ICH itself but an ongoing cognitive impairment process [70].

Pasi and colleagues [51] evaluated whether MRI-based cSVD burden assessment, in addition to clinical and CT data, improved the prediction of cognitive impairment after a spontaneous ICH [51]. Univariable and multivariable data analyses revealed an association between lobar CMB counts, lacunes, grade 3 periventricular WMHs (Fazekas scale), disseminated cSS, and cerebral atrophy scores. Furthermore, a higher risk of dementia was associated with WMHs and disseminated cSS after ICH. In particular, lacunes, severe DWMHs, and disseminated cSS were independently associated with post-ICH dementia risk after creating a multivariate model including all MRI markers, as well as other risk factors of interest. Finally, it is worth noting that MRI markers and scores for cSVD were found to be associated with rates of cognitive decline following ICH and capable of predicting the onset of post-ICH dementia—a clinical endpoint of immediate practical relevance for patients, families, and healthcare professionals [51].

Finally, it is vital to take into consideration a study by Fruhwirth and colleagues [50], who investigated whether the course of cognitive function in patients with recent small subcortical infarction was influenced by the severity of WMHs. Patients were assessed neuropsychologically using tests of global cognition, processing speed, attention, and set-shifting. Regarding the influence of deep WMHs on cognitive impairment, it was observed that all patients improved after 15 months, regardless of their WMH severity. Moreover, no differences were found between periventricular WMH groups at baseline and 15 months after the vascular event. It is essential to note that, in addition to demographics, when the potential influences of GM volume, the size of recent small subcortical infarctions, lacunes, microbleeds, and old cortical infarcts were controlled for, all models remained significant. However, a significant improvement in the prediction of attentional processes 15 months post-stroke was found from examining WMH volume, which explained a

substantial amount of variance (an increment of 17%). Finally, results from the statistical analyses showed that patients without or with only mild WMHs improved in set-shifting after 15 months. However, individuals with moderate-to-severe WMHs presented no improvement [50].

*4.3. Limitations*

Our systematic review is not without limitations. First, it is possible that different papers published by the same research groups reported findings for the same cohort. Moreover, the mediator and moderator roles of different sociodemographic characteristics (e.g., age, gender, education, occupational and/or marital status, annual and monthly income) in the association between LA and patients' clinical outcomes for mood and cognitive status have yet to be thoroughly studied. However, we provided a comprehensive presentation of different patients' sociodemographic and clinical characteristics that might enable indirect evaluation of such roles and provide additional valuable information for the implementation of future studies. Finally, despite the lack of consensus in the literature, our findings support the use of LA assessment as a valuable prognostic factor for depression and cognitive impairment among patients following ischemic stroke, thus indicating that a biomarker-based approach may provide important insights into the recovery potential of individual stroke survivors. Future quantitative studies (i.e., meta-analyses and mega-analyses) examining the prognostic role of LA in the treatment of PSD and PSDem/cognitive impairment are warranted, with the aim of providing more precise conclusions and guidelines regarding stroke patients' short-term and long-term care in cases of pre-existing LA.

**5. Conclusions**

Taking everything into consideration, the present review provides evidence for the prognostic significance of LA severity and its association with PSD and cognitive impairment/PSDem, as evaluated using baseline brain imaging and various post-stroke stages of different types of strokes.

Our findings suggest the significant role LA assessment could play in forecasting the development of conditions that could make an individual's quality of life poorer and harder. LA burden, serving as a surrogate marker of biological age and, consequently, "brain frailty" among stroke patients, appears to be able to yield additional information in terms of lesions that could potentially be involved in the development of post-stroke depression or responsible for an individual's cognitive dysfunction. Determining the extent of pre-existing WM abnormalities can properly guide decision making in an acute stroke setting. A greater degree of such lesioning is usually coupled with an unfavorable prognosis and poorer outcomes after implementing the selected intervention.

Additional data support the prognostic value of the presence of LA for the determination of the potential development of adverse outcomes, such as PSD or cognitive decline. WM lesions and WMHs were found to be significantly associated with the aforementioned symptoms post-stroke. Depending on age, sex, WMH volume, and the amount of time that has passed after a stroke, LA could predict or be highly related to the development of later adverse outcomes during the recovery stage. Nevertheless, LA assessment should be interpreted in a clinical context and may constitute a more powerful tool in conjunction with other clinical and neuroimaging biomarkers, facilitating individualized stroke care. Our review highlights the need for further investigation of LA as a promising imaging marker for clinical outcome that can enhance patients' risk stratification and facilitate overall stroke management. Additional studies with stroke individuals on the association between LA burden, depression, cognitive dysfunction, other adverse associated outcomes, and prognosis after stroke interventions are recommended in order to provide further insight into this clinically meaningful relationship and assistance for patients' short- and long-term personalized care.

**Supplementary Materials:** The following supporting information can be downloaded at: https://www.mdpi.com/article/10.3390/neurolint15010016/s1, Table S1: Characteristics of the included studies focusing on depression; Table S2: Characteristics of included studies focusing on cognition; Table S3: Characteristics of included studies focusing on depression and cognition.

**Author Contributions:** A.S. (Anastasia Sousanidou) and D.T. reviewed the literature, screened the reference lists, deleted duplicates and citations that did not meet the inclusion criteria, and assessed the articles; K.V. resolved any disagreements regarding the screening or selection process; T.K.D. and K.T. wrote the first draft of the manuscript; F.C., N.R., C.K. (Christos Konstantinidis), A.T., N.A., A.S. (Aspasia Serdari) and K.V. reviewed the tables, the presentation of the data, and the methodology. The corrected version was discussed collegially. E.T., F.C., C.K. (Christos Kokkotis) and S.K. wrote the final version of the manuscript. All authors have read and agreed to the published version of the manuscript.

**Funding:** This work was supported by the project "Study of the interrelationships between neuroimaging, neurophysiological and biomechanical biomarkers in stroke rehabilitation (NEURO-BIO-MECH in stroke rehab)" (MIS 5047286), which is implemented under the action "Support for Regional Excellence", funded by the Operational Program "Competitiveness, Entrepreneurship and Innovation" (NSRFm2014-2020), and co-financed by Greece and the European Union (European Regional Development Fund).

**Institutional Review Board Statement:** Not applicable.

**Informed Consent Statement:** Not applicable.

**Data Availability Statement:** All data discussed within this manuscript are available on PubMed.

**Conflicts of Interest:** The authors declare no conflict of interest.

## Abbreviations

| | |
|---|---|
| ACE | Addenbrooke's Cognitive Examination |
| AD | Alzheimer's disease |
| AIS | acute ischemic stroke |
| ARWMCS | Age-Related White Matter Change Scale |
| BMI | body mass index |
| cSVD | cerebral small vessel disease |
| CT | computed tomography |
| DES | executive dysfunction |
| DSM | Diagnostic and Statistical Manual of Mental Disorders |
| DWI | Diffusion-weighted imaging |
| DWMHs | deep white matter hyperintensities |
| FLAIR | fluid-attenuated inversion recovery |
| GM | gray matter |
| HDS-R | Revised Hasegawa Dementia Rating Scale |
| ICH | intracerebral hemorrhage |
| LA | leukoaraiosis |
| MCI | mild cognitive impairment |
| MDD | major depression disorder |
| MADRS | Montgomery–Åsberg Depression Rating Scale |
| MMSE | Mini-Mental State Examination |
| MRI | magnetic resonance imaging |
| NfL | neurofilament light chain |
| NIHSS | National Institutes of Health Stroke Scale |
| non-vaMCI | Non-vascular mild cognitive impairment |
| PRISMA | Preferred Reporting Items for Systematic Reviews and Meta-analyses |
| PSD | post-stroke depression |
| PSDem | post-stroke dementia |

| PVS | periventricular space |
| SVD | small vessel disease |
| T2WI | T2-weighted imaging |
| TIA | transient ischemic attack |
| TSH | thyroid-stimulating hormone |
| VCI | vascular cognitive impairment |
| WM | white matter |
| WMHs | white matter hyperintensities |
| WMLs | white matter lesions |

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
