# Peer review of "Leukoaraiosis as a Predictor of Depression and Cognitive Impairment among Stroke Survivors: A Systematic Review"

_2035-8377, doi:10.3390/neurolint15010016_

Round 1

Reviewer 1 Report

The manuscript by Tziaka et al. represents an attempt to summarize the literature data showing the leukoaraiosis as a predictor of depression and cognitive impairment among stroke survivors. While the authors performed a literature search selecting 34 suitable articles for the analysis, the manuscript misses any results and conclusions. All the data collected by authors is sunnitized within the Table 1, which is huge and uninformative. Further analysis of these data is needed to extract anything valuable.    

Reviewer 2 Report

1. Title should include "Systematic Review" to truly represent type of study

2. Abbreviations should be provided and explained below the tables

3. Tables are very difficult to read and not organized. 

4.  Quality assessment for each include study should be performed and provide data.

5. Who are “two independent investigators”? Need to clarify

6. Search terms in MEDLINE and Scopus are different. Please attach search terms that were used in each database as supplement for Data source and search strategies in the manuscript. Please provide details search terms in supplementary documents.

Reviewer 3 Report

Manuscript ID: neurolint-216004: Leukoaraiosis as a predictor of depression and cognitive impairment among stroke survivors

This is a comprehensive review, covering almost all the important clinical studies studying the value of leukoaraiosis as a surrogate marker for the post-stroke depression (PSD) and post-stroke dementia (PSDem). Although little statistical analysis was performed, the value of this report is not little. Because nowadays there are the problems with statistical analysis, the meaning of p-value has been discussed more and more. P-value does not tell everything. From this point of view, narrative studies like the present one should be more highly evaluated. This review will be a bibliographical encyclopedia to study the risk factors of PSD and PSDem. The manuscript is well-written and almost ready for publication. The following suggestions may help to increase further the value of this review.

(1) The authors' evaluation of each article in Table 1.

The important but difficult matter to read from the clinical studies is the evaluation of the biased factor of each study. Only the results often capture the readers, and the adequacy of the methods are less attended. From this point of view, the brief comments of the authors on each article may be added in Table 1. For example, the suggestions of excellent points or inferior ones of each article will be very helpful to evaluate each conclusion.

(2) The order of Table 1.

The order of the articles in Table 1 is not clear. Chronological order (new to old) may be common. It is related to the above comment (1), the best may be reliability order.

(3) The articles from the same group or institution should be specified in Table 1.

It is described as "4.3. Limitations Our systematic review is not without limitations. First, it is possible that different papers published by the same research groups reported findings on the same cohort." The articles from the same group or using the same cohort should be indicated in Table 1 or listed as a group.

(4) List of the abbreviations

The list of the abbreviations in the last or beginning of the text may be helpful.

End of File

Round 2

Reviewer 1 Report

I appreciate the authors for the great work making improvements. Nonetheless, I would like to ask them thinking a bit more on the optimization of the table 1. In my opinion, in the present form it is still huge and therefore uninformative but suitable in particular for supplementary materials if someone would like to analyze all the details. If there is any possibility to extract the most important data from the table 1  and present it in a more compact way?
